# Rose Flowers—A Delicate Perfume or a Natural Healer?

**DOI:** 10.3390/biom11010127

**Published:** 2021-01-19

**Authors:** Milka Mileva, Yana Ilieva, Gabriele Jovtchev, Svetla Gateva, Maya Margaritova Zaharieva, Almira Georgieva, Lyudmila Dimitrova, Ana Dobreva, Tsveta Angelova, Nelly Vilhelmova-Ilieva, Violeta Valcheva, Hristo Najdenski

**Affiliations:** 1The Stephan Angeloff Institute of Microbiology, Bulgarian Academy of Sciences, 26 Acad. G. Bontchev Str., 1113 Sofia, Bulgaria; illievayana@gmail.com (Y.I.); zaharieva26@gmail.com (M.M.Z.); almirageorgieva@gmail.com (A.G.); lus22@abv.bg (L.D.); nelivili@gmail.com (N.V.-I.); violeta_valcheva@mail.bg (V.V.); hnajdenski@abv.bg (H.N.); 2Institute of Biodiversity and Ecosystem Research, 2 Gagarin Str., 1113 Sofia, Bulgaria; gjovtchev@yahoo.de (G.J.); spetkova2002@yahoo.co.uk (S.G.); angelova_ts@abv.bg (T.A.); 3Institute of Neurobiology, Bulgarian Academy of Sciences, 21 Acad. G. Bontchev Str., 1113 Sofia, Bulgaria; 4Institute for Roses and Aromatic Plants, 49 Osvobojdenie Blvd, 6100 Kazanlak, Bulgaria; anadobreva@abv.bg

**Keywords:** *Rosa damascena* Mill., *Rosa alba* L., *Rosa centifolia* L., *Rosa gallica* L., essential oils, water and alcohol extracts, antimutagenic potential, antineoplastic effect, antiviral activity, antioxidant properties

## Abstract

Plants from the Rosacea family are rich in natural molecules with beneficial biological properties, and they are widely appreciated and used in the food industry, perfumery, and cosmetics. In this review, we are considering *Rosa damascena* Mill., *Rosa alba* L., *Rosa centifolia* L., and *Rosa gallica* L. as raw materials important for producing commercial products, analyzing and comparing the main biological activities of their essential oils, hydrolates, and extracts. A literature search was performed to find materials describing (i) botanical characteristics; (ii) the phytochemical profile; and (iii) biological properties of the essential oil sand extracts of these so called “old roses” that are cultivated in Bulgaria, Turkey, India, and the Middle East. The information used is from databases PubMed, Science Direct, and Google Scholar. Roses have beneficial healing properties due to their richness of beneficial components, the secondary metabolites as flavonoids (e.g., flavones, flavonols, anthocyanins), fragrant components (essential oils, e.g., monoterpenes, sesquiterpenes), and hydrolysable and condensed tannins. Rose essential oils and extracts with their therapeutic properties—as respiratory antiseptics, anti-inflammatories, mucolytics, expectorants, decongestants, and antioxidants—are able to act as symptomatic prophylactics and drugs, and in this way alleviate dramatic sufferings during severe diseases.

## 1. Introduction

The beauty and fragrance of rose flowers have been known since ancient times. The first historical reports of roses were found in old Chinese and Sanskrit texts. Fossil findings dating from about 40 million years ago indicate that *Rosa* species have existed on the planet since at least that time [1,2,3]. Up to the present day, roses are one of the most important groups of ornamental plants, a sign of inspiration, purity, love, happiness, and beauty, called the “Gift of angles”, “Queen of flowers”, and “Gol-E-Mohammadi” [1,2,3,4].

According to the findings of Nazarenko et al. (1983), the genus Rosa, to which the oil-bearing species belongs, originates from the ancient evergreen lianas of the Sundarbans in India, called “the pharmacy of the world” as more than a quarter of the drugs known today in medicine are based on plants from these forests [5]. At present, about 1000 genotypes of roses are known; they are classified and grouped based on botanical characteristics: hybrid teas, grand floras, polyanthus, floribundas, miniatures, climbing, shrub, but only a few of them exhibit the marked fragrance which is preferred by perfumers [6,7]. *Rosa damascena* Mill. forma *trigintipetala* Dieck, *Rosa alba* L. *Rosa damascena* Mill. var. alba, *Rosa gallica* L. *Rosa centifolia*, *Rosa chinensis*, and *Rosa rugosa* are grown worldwide predominantly as a raw material for the perfume and cosmetics industry [1,8,9,10,11,12,13]. Species affiliation is a determining factor of the quality of commercial products: essential oils, hydrolates, and the concrete and absolute. Essential oils and hydrolates are products of distillation, the concrete is an extract with a non-polar solvent, and the absolute is a subsequent product which is formed after the extraction of concrete with ethanol [12]. Among the world leaders in rose production are countries such as Turkey, China, countries of the former Soviet Union, Egypt, Morocco, and Bulgaria [13,14,15]. In Bulgaria, the cultivation and processing of roses is a tradition and the livelihood of a large part of the population and it is very important for the country’s agricultural economy, and *R. damascena* Mill., *R. alba* L., *R. gallica* L., and *R. centifolia* L. are grown predominantly [15,16]. The use of rose essential oils, aqueous and alcoholic extracts of rose petals, flowers, and heads, not only as fragrances or for aromatherapy, but also for the treatment and prevention of various diseases and disorders is very popular in folk medicine. Thus, the search for new biological activities of plant extracts is in many cases based on data from recipes from folk healers. On a scientific basis, the empirical knowledge from folk medicine is an important aspect of more and more in vitro and in vivo studies, including preclinical and clinical trials. These trials explore and explain the therapeutic efficacy of rose products and their ingredients, including: antidepressant effects, psychological relaxation, improvement of sexual dysfunction, antioxidant, antimicrobial, antifungal, probiotic and antipyretic effects, smooth muscle relaxation, lipid-lowering content, antiulcerogenic effects, etc. [2,8,17,18,19,20,21,22]. Rose oils are recommended not only for inhalation and topical application (in aromatherapy and dermatology) but also for oral administration at physiologically applicable doses. A number of studies have shown the anticancer activity of rose oils and their potential to be used as adjuncts in adjuvant therapy of tumors [23,24]. Rose oil is also known for its anti-HIV, antibacterial, and antioxidant properties [3,6,25,26,27]. In this review, we are considering *R. damascena* Mill., *R. alba* L., *R. centifolia* L., and *R. gallica* L. as raw materials important for producing commercial aromatic products, analyzing and comparing the main biological activities of their essential oils, hydrolates, and extracts as natural healers. For this purpose, information from the databases PubMed, Science Direct, and Google Scholar was used.

## 2. A Short Introduction on Habitat, Cultivation, and Comparative Morphological Characteristics of Rose Plants

The cultivation of roses has been widespread in temperate climates throughout the world. They are best known for their beauty as ornamental flowers, but roses are an industrial product as well. Most of the rose species are native to Asia, (Middle Eastern locations including Damascus, Hama, Aleppo, Homs in Syria; Iran). Roses are cultivated in Europe (France, Italy, Central Bulgaria at the Rose Valley of Kazanlak, Turkey, and Russia), North America, and Northwest Africa (Morocco), and in China, Japan, Korea, and India (Kashmir, Bihar, Uttar Pradesh, and Punjab States) (Figure 1). Other producers are Afghanistan and Chile [28,29]. At the beginning of the 19th century, due to the growing needs of the perfume industry in Western Europe, attempts were made to introduce *R. damascena* Mill. in countries such as France and Germany [9], and later in the Crimea, Ukraine, but it failed to adapt to the conditions of the Crimean Peninsula [5]. Presently, some oil-bearing *Rosa* species are cultivated for commercial purposes in the Middle East, Iran, Turkey, and Bulgaria. Rose oil, hydrosol, concrete, and absolute are widely appreciated and used in the perfumery, pharmacy, and food industries. The biggest producers of these products in the world, supplying more than 90% of the total rose oil, are Bulgaria, Turkey, and Iran [30], but the best quality oil is Bulgarian *R. damascena* oil [4].

### 2.1. Rosa Damascena Mill. Forma Trigintipetala Dieck

*R. damascena* Mill., i.e., damask rose with thirty corolla leaves, is the most important scented rose species that originated in Central Asia. According to Iwata et al. (2000), the oldest damask roses have a three-parent origin—*R. moschata*, *R. gallica*, and *R. fedschenkoana* [31]. Many scientists have worked on the genesis of the Bulgarian rose. Some authors believe that it is a cross between *R. gallica* L. and *R. canina* L., others suggest *R. gallica* and *R. moschata*, *R. centifolia* and *R. alba*, or *R. centifolia* and *R. gallica* [14]. Topalov (1978) defines the Kazanlak oil-bearing rose as an independent species—*R. kazanlika* V. Top. Sp. Nova Sect. Gallicanae D.C. The taxonomy of the species according to the Germplast Resources Information Network (GRIN) [32] cited *Rosa X damascena* Mill. (=*R. gallica* X *R. moschata*). The botanical characteristics of *R. damascena* can be described as follows: it is a perennial shrub, strongly branched, and reaching a height of 2.5 m. The branches are light green with large, curved down flattened spines. The leaves have five to seven elliptical toothed leaf parts. The flowers are collected in panicle inflorescences with 3–9 flowers (sometimes up to 31), which bloom consecutively. They have 5–60 corolla petals, colored pale pink to pinkish red [29].

### 2.2. Rosa alba L.—Rosa damascena Mill. var. alba

*R. alba* L. (or *R. damascena* Mill. var. alba) is known as the Bulgarian white rose. As an ornamental plant, it has a long history in Bulgarian folklore, holy beliefs, wedding rituals, and religious traditions [33]. Although scarce, information on the origin, distribution, biology, plantations, and white rose processing has always been present together with that of *R. damascena*. As one of the old roses, it is thought to have originated in the Mediterranean or the Middle East and is considered as a parental form of *R. damascena*. In the past, it was grown and processed mixed with *R. damascena* Mill. for industrial purposes. Currently, the white rose is cultivated separately in the Roses’ Valley in Bulgaria only, and the essential oil of *R. alba* L. is distilled separately too [34]. *R. alba* L. is found also in the central west region in the state of Goiás, Brazil, and some areas of Turkey, and is popularly known as the white rose, yard rose, backyard rose, white rose of York, or Sufaid Gulab [35].

### 2.3. Rosa gallica L.—Gallic or Red Oil-Bearing Rose

This rose occurs in the wild in the latitudes of the Middle East and Asia Minor, the Caucasus, Crimea, as well as in Southeastern and Central Europe. Due to its widespread use in France, the first European country to adopt this type of rose from the Middle East, it was called “French” or “Provencal”. It is one of the ancient oil-bearing roses, mentioned as the parent of almost all species in the Gallicanae section. Reaching Russia and Central Asia, today it is the main species grown in these lands. Eastern nations use it mainly for its dried flowers and buds [32].

### 2.4. Rosa centifolia L.—A Rose with a Hundred Leaves

This rose has been known since ancient times and is still found in the wild. It is used for both decorative and industrial purposes. It is considered to be a complex hybrid between *R. gallica*, *R. moschata*, *R. canina*, and *R.damascena* [31].

As the content of essential oil in its flowers is very low (three times less than that of *R. damascena*), *R. centifolia* L. is used mainly for the production of concrete and absolute (France and Morocco, Italy) [36], dried fruits with high therapeutic value and content of vitamin C, dried flowers, and rose water. *R. centifolia* L. is the subject of research due to its nutritional qualities as the petals contain ingredients with an antioxidant effect and good organoleptic properties and microbial parameters. The fruits are rich in sugars, organic acids, and potassium, all of which are ingredients that preserve product quality and have advantages in technological processes (syrups, juices, jam). In Pakistan, there is research work done which is aimed at developing expensive aromatic rose products, focusing on the extraction of *R. centifolia* [13].

## 3. Composition of Rose Oils

The predominant rose species used worldwide for the production of rose oil is the thirty-petalled *R. damascena* Mill. forma *trigintipetala* Dieck. In addition, *R. gallica* L., *R. centifolia* L., *R.alba* L., and *R. rugosa* L. are grown worldwide [11,12]. The yield and composition of rose oil obtained from the rose flowers have been known to be strongly affected by the geographic area, the soil, the climate, the moment of the picking, the storage conditions, and the technology parameters. The genotype is decisive for the chemical profile when all other conditions are constant.

Perfumery is the main consumer of rose oil. It is a product of double distillation and its composition varies within certain limits. The international standard ISO 9842 sets limits on the main ingredients for rose oil from *R. damascena* Mill. [37]. Due to the complexity of the composition, the standard is a necessary but insufficient quality criterion and serves rather as a rough guide in characterizing the oil. Chemically, it is a mixture of almost 300 compounds such as terpene and phenol derivatives of hydrocarbon compounds [38,39]. A small number of sulfur-containing components have also been identified [40].

Table 1 and Figure 2 show the typical composition of the essential oils of the main types of roses grown on industrial farms in Bulgaria [41]

*R. damascena* oil is characterized by a high content of terpene alcohols, the majority of which is citronellol, followed by geraniol, nerol, and linalool. The relationship between them is also important. Oils with a citronellol/geraniol ratio of 2.5:4.3 [63] or 1.25:1.30 [64] are considered of high quality. The main hydrocarbons have a either a C_17_, C_19_, or C_21_ skeleton [39].

In the case of the white rose, different dynamics are observed in the ingredients, from a typical citronellol [65] to a categorically geraniol model [16]. This is probably due to variability in the population itself [66]. In summary, the level of terpene alcohols approaches that of *R. damascena* Mill. Geraniol is known to carry pharmacological effects in rose oil [67,68]. A low content of methyleugenol and a high content of geraniol guarantees safe application with proven biological effects of *R. alba* L. essential oil.

In *R. gallica* L. the oil has a typical geraniol pattern [16]. The level of hydrocarbons is also very high. As a rule, the content of methyleugenol is low. This rose is widely grown in the countries of the former Soviet Union. The processing technology in these areas has other traditions and for this reason the composition of the oil is often quite different—phenylethyl alcohol is the major component [5].

In *R. centifolia* L., the total content of terpene alcohols is low [16]. The oil profile is typically geraniol. This species also produces low levels of methyleugenol. The high values of paraffins make it undesirable for perfumery purposes, but in practice it is used to obtain rose water and extracts. 

Unlike *R. damascena*, the other three roses have hydrocarbons that are shifted to the following homologues: C_19_, C_21_, and C_23._

The quality of both types of essential oils cannot be compared with that of *R. damascena* Mill. and *R. alba* L.

Rose water is a by-product in the distillation of rose oil or is obtained by using less raw material. It contains a minimum amount of oil (about 0.02–0.09%), the composition of which differs from regular oil in its high content of phenylethyl alcohol [69,70].

## 4. Biological Activities of *Rosa* Species

### 4.1. Cytotoxic/Anticytotoxic, Mutagenic/Antimutagenic and Genotoxic/Anti-Genotoxic Potential

The study of the cytotoxic/anticytotoxic, mutagenic/antimutagenic, and genotoxic/anti-genotoxic potentials of the extracts and oils obtained from plants of the Rosaceae family is of particular importance for the protection of the cell genome from damage induced by various factors (Table 2). The selection of concentrations that are harmless and do not damage normal cells is particularly important in therapeutic approaches in the fight against cancer. 

Various test methods were used to study the toxic/cytotoxic, mutagenic/antimutagenic, genotoxic/anti-genotoxic potential of the rose extracts and oils, both in vivo and in vitro. The MTT test, BrdU assay, double-staining fluorescence assay, and TUNEL (Terminal deoxynucleotidyl transferase dUTP nick end labeling assay) give information about the toxic/cytotoxic activity. They assess the cell viability, proliferative activity, and induction of apoptosis. The reverse mutation, somatic mutation, and the recombination test (SMART) are applied to study the mutagenic/antimutagenic activity, and to detect the genotoxic/anti-genotoxic potential. The tests for the induction of chromosomal aberrations, micronuclei, as well as comet assay, are used. They assess the frequency of mutations, changes in DNA integrity, changes in chromosome structure, and abnormalities in the course of mitosis.

With regards to *R. damascena* extracts and oil, *R. damascena* Mill. aqueous, aqueous-alcoholic, ethanol, and methanol extracts were tested (Table 2). The cytotoxic activity of these extracts was found to be directly dependent on the type of extraction, the concentration applied, and the cell sensitivity. Normal animal and human cells, as well as different cancer cell lines, show different sensitivity to aqueous extracts of *R. damascena* [71]. The aqueous extract possesses cytotoxic activity against human HeLa tumor cell line NCBI and human lymphocytes with increasing concentrations assessed by MTT analysis (Test for cell viability as a function of redox potential). The IC_50_ for cancer cell lines is 0.0045 and for lymphocytes 115.7 mg/mL, respectively. The highest concentration (0.5 mg/mL) reduces the viability of HeLa cells and lymphocytes by 94.72% and 47.08%, respectively. The aqueous and ethanol extracts of *R. damascena* Mill. reduce the viability and proliferative activity of human gastrointestinal cell lines (AGS) and are able to induce apoptosis in a concentration-dependent manner [72]. The aqueous extract shows a lower inhibitory effect than the alcoholic one, probably due to its different composition. The cytotoxic effect of the alcoholic extract increases with the incubation time in HeLa cells [73]. The IC_50_ values for this cell line when treated with the alcoholic extract are 2135, 1540, and 305.1 μg after 24, 48, and 72 h, respectively. 

The rose extracts do not show or have low genotoxic activity at the concentration range tested. Alcoholic extracts (1%) of the *R. damascena* of Bulgaria do not induce DNA damage in human lymphocytes in vitro, assessed by comet assay [74].

The essential oil possesses cytotoxic and genotoxic activity, clearly dependent on the concentration applied and the cell type treated (Table 2). The oil of *R. damascena* Mill.from Iran is safe at a low dose (10 μg/mL) both in normal NIH3T3 and cancer cells (A549), but the higher concentrations in the range of 50–200 μg/mL increase cytotoxic and genotoxic effects, assessed by micronucleus assay [24]. A549 cells show a lower IC_50_, indicating their higher sensitivity to rose oil than the normal fibroblast cellsNIH3T3. Concrete and absolute oil of *R. damascena* var. *trigintipetala* Dieck exhibited cytotoxic activity against cancer cell lines HepG2 and MCF7, where IC_50_ for the concrete was 16.28 and 18.09 μg/mL and for absolute was 24.94 and 19.69 μg/mL, respectively [75]. The authors reported that both rose oils are cytotoxically and genotoxically safe in normal human lymphocytes at concentrations of 5 and 10 μg/mL. *R. damascena* oil showed clear dose-dependent cytotoxic activity in cancer cell lines A549, PC-3, and MCF-7, as well as antibacterial effect in *P. acnes* [76]. The essential oil of *R. damascena* (1%) induced a lethal effect in human lymphocytes in vitro after incubation for 1 h. The same effect was observed with 0.1% after 24 h [74]. 

In the last decade, some studies have shown the cytoprotective and genoprotective potential of the extracts and oils of *R. damascena* [74,75]. Water and alcoholic rose extracts were tested for their antimutagenic activity. Alcoholic extracts (1%) of the *R. damascena* of Bulgaria and *R. centifolia* have genoprotective effects against 25 μmol/L of hydrogen peroxide in human lymphocytes [74]. Limited data exist on the cytoprotective and genoprotective potentials of the roseoil. Hagag et al. (2014) found that concrete and absolute (10 μg/mL) of *R. damascena* var. *trigintipetala* Dieck exerted a strong antimutagenic effect against the chemotherapeutic agent mitomycin C-induced damage in normal human lymphocytes [75]. They enhance the proliferative activity and decrease the mutagen-induced chromosome aberrations. 

There is scarce data on the antiproliferative effect of *R.centifolia* extracts and oil. Kalemba-Drożdż and Ciermiak (2019) found that the aqueous-alcoholic extracts of this species (1% tinctures diluted 100-fold in RPMI medium) did not affect the viability of human lymphocytes when applied for 1 h and 24 h. Rose extracts do not show or have low genotoxic activity at the tested concentrations [74], whereas the tested oil of *R. centifolia* has cytotoxic activity. The essential oil of *R. centifolia* in a concentration of 1% showed a high toxic effect in human lymphocytes in vitro after incubation for 1 h. 

Limited studies exist regarding the antimutagenic potential of the rose extracts. Aqueous extracts of three different cultivars of *R. centifolia* petals (1.5 mg/mL) reduce the frequency of mutations (from 4% to 55%) in *E. coli* cells compared to those induced by the direct mutagen ethyl methanesulfonate [79]. It is interesting to note that the red colored cultivar “passion” shows the highest antimutagenic ability. 

With regards to *R. alba*, the cytotoxicity and genotoxicity of these extracts has been identified in the literature. The essential oil of Bulgarian *R. alba* L. when tested in concentrations of 250, 500, and 1000 μg/mL exhibited a weak cytotoxic and genotoxic effect in plant cells of *H. vulgare* [77,78]. No significant increase is observed for the level of mitotic disturbances (micronuclei and aneuploidy effects), as well as no significant effect on the induction of chromatid aberrations, compared to the untreated control.

The *R. alba* essential oil possesses genoprotective potential. Data of Gateva et al. (2020) suggest a promising pharmacological potential of Bulgarian white rose essential oil applied in non-toxic concentrations in barley root tips (250, 500 μg/mL) and in human lymphocytes in vitro (50, 200 μg/mL), owing to a well-expressed anticytotoxic/anti-genotoxic potential against the direct alkylating agent N-methyl-N′-nitro-N-nitrosoguanidine (50 μg/mL). This effect was manifested by increasing the cells’ proliferation and decreasing both chromosome aberrations and micronuclei, regardless of the experimental schemes used, when applied in nontoxic concentrations [77].

With regards to *R. gallica*, thus far there is no information available on the cytotoxic/genotoxic as well as anticytotoxic/anti-genotoxic effects of these oils or extracts, either in the field of basic research or from clinical studies [80].

Briefly, the analyzed extracts and oils of these roses demonstrated an effective cytotoxic effect on cancer cells and a weak effect on normal cells when applied at low concentrations. The rose extracts show weak genotoxic activity and do not cause DNA injuries, whereas the oils possess significant concentration-dependent and/or individual genotoxic activity. Both rose extracts and essential oils possess well-expressed antimutagenic and anti-genotoxic potentials against different chemical mutagens. In this way, the oils and extracts of these representatives of the Rosaceae family could be used as promising chemotherapeutic agents in the treatment of cancer, to deal with the secondary neoplasmic processes resulting from chemotherapy.

### 4.2. Antiviral Activity of Representatives of the Rose Family and Their Metabolites

The representatives of the Rosaceae family are of great economic importance to humanity. Most of them are especially valuable for their taste and are used as food, as spices, or in the form of tea. Some have been used in traditional medicine in many countries for thousands of years, while others are used in the cosmetic industry for various creams, perfumes, and other products.

The antiviral activity of members of this family has not been well studied and information about their impact on viral replication and the mechanism by which this impact is produced is scarce. The research carried out by different teams in this area opens up many perspectives for the study of different products obtained from different vegetative and propagating parts of these precious plant species.

Antiviral activity against enteric coronavirus is exhibited by extracts of two representatives of the Rosaceae family, *Rosa nutkunu* and *Amelunchier alnifoliu*, and the respiratory syncytial virus is completely inhibited by *Potentillu urgutu* root extracts [81]. Methanolic extract of the *Rosa macrophylla* flower shows an inhibition of influenza A/WSN/33 (H_1_N_1_) virus replication [82].

Twelve extracts of plants belonging to the Rosaceae family were tested for inhibitory activity against HIV-1 protease. Of all the extracts tested, the strongest inhibitory effect was shown by the *R. rugosa* root extract and the *Prunus sargentii* leaf extract, at concentrations of 100 µg/mL [83]. The effect of the aqueous and methanol extracts of *R. damascena* on HIV infection in vitro was investigated. The activity of the crude extract is thought to be the result of the synergistic action of various compounds whose combination acts at different stages of replication of the virus [84,85].

Organic products that are extracted from different plant parts are an endless source of molecules with therapeutic potential. Many of the secondary metabolites isolated from plants exhibit antiviral activity that can be expressed in impaired viral reproduction within the host cell at different stages of viral replication or inactivation of extracellular virions.

Some of the products found in rose all, present in the highest percentages, are different types of terpenes. In vitro antiviral activity of citral has been demonstrated to block yellow fever virus replication [86]. Citral also inhibits infection with non-enveloped murine norovirus (MNV), the action being time-dependent and most likely due to nonspecific interactions with extracellular viral particles, which prevents its attachment to sensitive cells [87].

Citronellol and eugenol compounds are very active against the influenza virus after only 10 mins ofexposure [88]. The common components in rose oil, including nerol, citral, citronellal, citronellol, geraniol, and eugenol, show antiviral activity against herpes simplex virus type-1 (HSV-1) and parainfluenza virus type-3 (PI-3) [89]. Geraniol also shows low antiviral activity against coxsacki virus B1 (CVB1) replication with a selective index of 3.9 [90].

Several substances that effectively inhibit HIV were isolated, such a rosamultin, isolated from the root of *R. rugosa*, which inhibits by 53% the HIV-1 protease at a concentration of 100 µM. Nine compounds isolated from *R. damascena* methanol extract showed anti-HIV activity, with a different mechanism of action being established. Tetrahydroxyflavanone (kaempferol 1) selectively inhibits viral protease, and two 3-substituted kaempferol derivatives as well as pentahydroxyflavone (quercetin, 2) inhibiting HIV infection by preventing the binding of gp120 to CD4. Irreversible interaction with gp120 also produces 2-phenylethanol-O- (6-O-galloyl) -beta-D-glucopyranoside [84,85].

The increasing research of the Rosaceae family regarding their impact on the replication of different kinds of viral families could lead to their natural use as antiviral agents. Furthermore, during a serious epidemic situation, such as COVID-19, people have faced severe challenges and both physical and emotional stress. Rose essential oils with their valuable therapeutic properties—as respiratory antiseptics, anti-inflammatories, mucolytics, antitussives, expectorants, decongestants, and antioxidants—are able to act as symptomatic drugs and potentially help during a COVID-19 infection [91].

### 4.3. Antimicrobial Activities

According to the Interagency Coordinating Group (IACG) on Antimicrobial Resistance (AMR), drug-resistant diseases could cause 10 million deaths each year by 2050. Nowadays, at least 700,000 people die each year due to drug-resistant bacterial infections, including multidrug-resistant tuberculosis, of the respiratory and urinary tract, and sexually transmitted infections [81]. The significant number of studies on the use of essential oils and their components against multidrug-resistant bacteria show the exceptional potential of these natural products to curb the development of antibacterial resistance.

Table 3 shows the highest antimicrobial activities of essential oils and extracts of the observed *Rosa* species. It is known that the antimicrobial effects of the studied essential oils are dependent to a large extent on their metabolic profile. The antimicrobial activity of the rose essential oils can be attributed to the contained monoterpenes that, due to their lipophilic character, act by disrupting the microbial cytoplasmic membrane, which thus loses its high impermeability for protons and larger ions [8,27,92]. As antimicrobial agents, essential oils are able to target the bacterial cell walls and membranes, damage their permeability, and cause the collapse of normal cellular processes. When a disturbance of membrane integrity occurs, its functions are compromised.

Andoğan et al. (2002) found citronellol (10.3–46.7%), geraniol (2.8–23.3%), nerol (1.3–11.9%), and linalool (0.6–0.8%) to be main components in *R. damascena* Mill. essential oil of Turkish origin. They observed the antimicrobial activities of some of the individual substances and proved that citronellol, geraniol, and nerol have more potent antimicrobial activity against *Staphylococcus aureus* individually (with inhibition zones of 20 mm, 21 mm, and 19 mm, respectively) than in a mixed form (8 mm). The essential oil was not active against *Escherichia coli* and *Pseudomonas aeruginosa* [97]. Özkan et al. (2004) investigated its antimicrobial activity by the agar diffusion method of two methanol (MeOH) extracts: from fresh flowers and spent flowers collected in Turkey at different concentrations (10%, 5%, 2.5%, and 1%). The extracts from the fresh flower (10%) showed a low effect against *Aeromonas hydrophila*, *P. fluorescens*, and *S. aureus*, as did the spent flower at the same concentration against *Bacillus cereus*, *Enterococcus feacalis*, *P. aeruginosa*, *Salmonella typhimurium*, and *Yersinia. enterocolitica.* Both extracts exhibited low antimicrobial activity against *Enterobacter aerogenes* and *Klebsiella pneumoniae* and were not active against *E. coli*. The 5% MeOH extract from the fresh flower showed a low effect against all tested microorganisms, except *E. coli*, and the extract from the spent flower did the same except for *A. hydrophila* and *Mycobacterium smegmatis*. Both 2.5% and 1% extracts from the fresh and spent flowers showed low effects or were not active against all tested microorganisms including *Proteus vulgaris* and *S. enteritidis* [27]. Gochev et al. (2008) compared *R. damascena* Mill. oils of Bulgarian, Moroccan, Chinese, Turkish, and Iranian origin. The Bulgarian oil showed the highest antimicrobial activity against all tested microorganisms: *B. cereus*, *S. aureus*, *S. epidermidis*, *E. coli*, and *C. albicans*, followed by the essential oil of Turkish origin. Essential oils from Morocco and China showed weaker effects compared to oils from Turkey and Iran against all strains studied. The pure substances geraniol, nerol, and linalool showed the same activities as the Bulgarian rose oil against Gram-positive bacterial strains (the minimal inhibitory and bactericidal concentrations (MIC/MBC) varying between 128 and 256 µg/mL), *E. coli* (MIC = 512 µg/mL and MBC = 1024 µg/mL), and the tested fungi from the type *Candida* (MIC = 1024 µg/mL and MBC = 2048 µg/mL). Citronellol was less effective (MIC = MBC from 256 to 512 µg/mL). In addition, the phenylpropanoid eugenol was characterized with the highest antimicrobial activity against all tested Gram-positive bacteria, including *E. coli* (with MIC = MBC between 64 and 128 µg/mL), as well as against *P. aeruginosa*, *P. fluorescens* (food spoilage strain, isolated from minced meat), *C. albicans*, and clinical isolates of the testfungi (with MIC = MBC from 512 to 1048 µg/mL) [93]. Essential oils from *R. damascena* Mill. and *R. alba* L. of Bulgarian origin showed low antifungal activities against *A. niger* and *A. flavus* [20].

Bulgaria is a major producer of oil from *R. alba* L., thus many Bulgarian scientists focus their work on studying the biological activities of this oil [8]. Mileva et al. (2014) investigated the chemical composition and antimicrobial effect of white rose essential oil and its ingredients originating from the Valley of Roses, Kazanlak. The authors compared the MIC of *R. alba* L. oil with the MIC of nerol, eugenol, citronellol, methyleugenol, and geraniol. Interestingly, the oral pathogen *A. actinomycetemcomitans* proved to be the most sensitive bacterium to eugenol and citronellol (with MIC = 0.17 mg/mL), as well as to other pure compounds—about 0.2 mg/mL—with the exception of geraniol (MIC = 0.37 mg/mL) and *R. alba* L. oil 0.45 mg/mL. *S. mutans* is sensitive to eugenol (MIC = 0.17 mg/mL), followed by equal MICs of nerol and rose oil (0.82 mg/mL), geraniol (MIC = 1.12 mg/mL), methyl eugenol (MIC = 1.19 mg/mL), and citronellol (MIC = 1.74 mg/mL). *E. faecalis* is more resistant to nerol and oil than *R. alba* L. (MIC = 1.36 mg/mL). Geraniol and methyl eugenol in opposition to enterococci showed the same effect as against *S. mutans* (MICs were 1.12 mg/mL and 1.19 mg/mL, respectively). Citronellol had higher anti-enterococcal activity with a MIC of 0.69 mg/mL [21]. Another study of the rose oil produced from petals of *R. alba* L. showed that geraniol was characterized by the lowest antifungal activity, followed by nerol, citronellol, methyl eugenol, and eugenol against *A. niger* and *A. flavus* [20]. Carvalho et al. (2008) showed the weak antimicrobial effects of aqueous extract from *R. alba* L. of Brazilian origin against *S. aureus*, *E. coli*, and *C. albicans* (inhibition growth zone 9 mm) [98]. Gochev et al. (2010) found, through thedisc diffusion method, theweak antimicrobial activity of Bulgarian essential oil from *R. alba* L. against *E. coli*, *S. aureus*, *S. epidermidis*, *S. abony*, *C. albicans*, and *C. tropicalis* (10–14 mm inhibition zone) and no effect against *P. aeruginosa*, *P. fluorescens*, and *P. putida* [8]. In addition, these authors examined the MIC and MBC of essential oil, including the pure compounds citronellol, geraniol, and nerol. The individual substances geraniol and nerol exhibited highest antimicrobial activity against all tested strains, with MIC varying between 0.01% and 0.2% and MBC ranging from 0.01% to 0.41% (*w/v*). Citronellol ranks in a middle position with MIC from 0.03% to 0.41% and MBC between 0.03% and 0.82% (*w/v*), followed by rose essential oil with MIC varying from 0.05% to 0.82% and MBC from 0.05% to 1.64% (*w/v*) against all microorganisms [8].

According to the literature data, *R. centifolia* showed moderate activities against Gram-positive and Gram-negative bacteria [95,99]. The crude ethanol (EtOH) extract from *R. centifolia* L. of Palestine origin showed a weak antibacterial effect both against Gram-positive (*S. aureus*) and against Gram-negative (*K. pneumonia*, *P. vulgaris*, and *P. aeruginosa*) bacteria [95]. The ethanol extracts of this plant showed moderate activities against *S. aureus*, *K. pneumoniae*, *E. cloacae*, and *Acinetobacter baumannii* and no effect against *E. coli* and *E. faecalis* [94]. Gauniyal and Teotia (2015) also reported the moderate antimicrobial activities of EtOH extracts from *R. centifolia* L. against *Lactobacillus acidophilus* and *C. tropicalis*, and no activity against the Gram-negativespecies *E. faecalis* [96].

Mileva et al. (2014) investigated the major compounds in essential oil of *R. gallica* L. originating from Moldova. The main compounds found were citronellol 10.29%, *n*-nonadecane 8.79%, geraniol 8.56%, and nerol 4.18%. Essential oil of this rose showed very low antifungal activities [20]. In general, the antimicrobial potential of *R. gallica* is less studied. According to Bonjar (2004), the methanol extract of *R. gallica* showed weak antibacterial effects against the Gram-positive bacteria (*S. aureus* and *B. cereus*) with an inhibition growth zone around 7–9 mm, except for *S. epidermidis*, *B. pumilus*, and *Micrococcus luteus*, which were resistant, as well as against the Gram-negative bacterial species *E. coli*, *P. aeruginosa*, *P. fluorescens*, *K. pneumoniae*, *Bordetella bronchiseptica*, and *Serratia marcescens* [100]. The essential oils from *R. gallica* L. of Iranian and Moldovan origin showed a weak antimicrobial effect against *S. saprophyticus*, *S. typhi*, *Shigella flexneri*, *A. niger*, and *A. flavus*, and no effect against *S. aureus*, *S. epidermidis*, and *E. coli* [20,101].

### 4.4. The Role of Rose Oils in the Treatment of Respiratory Tract Diseases

As can be seen, many research groups have focused their research programs on investigating the antimicrobial activities of plants from the *Rosa* species, and their extracts, in the hope of discovering new antibacterial agents. The antimicrobial potency of rose ethanolic extract and essential oils against respiratory tract pathogens has been investigated for many years. The application manner of this oil is variable, depending on the symptoms of the disease and the treated area. The commonly used method is the inhalation of rose oil, which significantly decreases oxyhemoglobin concentration and activity in the right prefrontal brain cortex and increases feelings of comfort. Haze et al. (2002) found that the vaporous inhalation of rose oil decreases relative sympathetic activity of the nervous system, as measured by heart rate variability and low frequency amplitudes of systolic blood pressure in healthy adult females [102]. It has been reported that rose essential oil also can be inhaled after sprinkling a drop or two of the oil onto a cloth or tissue. Inhaling essential oil molecules, or absorbing essential oils through the skin, transmits messages to the limbic system—a brain region responsible for controlling emotions and influencing the nervous system. The inhalation of rose oil showed protective effects against damages caused by intoxication with formaldehyde in the male reproductive system [103,104]. Rose oil also enhanced ileum contractions and gastrointestinal motility in rats [49]. There are data reporting that the main rose oil components, citronellol, geraniol, and nerol, had strong antimicrobial activity against some bacteria. Asthma is a disorder that is caused by chronic inflammation in the airway. One of the major symptoms of asthma is a chronic cough. Although there is a large number of research which indicates some progress in asthma treatment, local and systematic adverse effects of inhaled corticosteroids can lead to alternative and complementary remedies. Herbal remedies, because of their effective constituents, have been accepted as a third effective treatment in asthma [105]. Investigations regarding the effects of herbal remedies in asthma treatments are still inconclusive and there is evidence regarding the benefits of plant essential oils in the treatment of asthma. It is accepted that there is direct correlation between bronchodilating potency and cough inhibition [106]. It has been elucidated that polyphenols and flavonoids have a significant effect on cyclooxygenase (COX)-1, (COX)-2, 5-lipoxygenase (5-LOX), and 12-LOX, due to their anti-inflammatory properties. It seems that the anti-inflammatory effects might be useful in cases of asthma where pathogenesis inflammation plays a major role. The above-mentioned evidence suggests that there is a close correlation between asthma, inflammation, and oxidative stress. Essential oils are a good choice of treatment that can help millions of people affected by asthma [107].

### 4.5. Free Radical Scavenger and Antioxidant Activity of Rose Extracts

Oxygen is a very important molecule for aerobic life, necessary for respiration and energy metabolism, but it may also be dangerous in some conditions due to its ability to form reactive oxygen species (ROSs). Normally, cells control the levels of ROSs by balancing the generation of ROSs with their elimination by antioxidant defense systems. Under extreme conditions, this balance is disturbed, and over-generated ROSs are capable of oxidizing cellular proteins, nucleic acids, and lipids, thus violating their function [108].

ROSs contribute to many diseases as cellular aging and mutagenesis, with a mechanism of lipid peroxidation, which cause the destabilization of cell membranes, DNA damage, and the oxidation of low-density lipoprotein (LDL) [109].

The antioxidant power of biomolecules is an expression of their capability to defend from the actions of the free radicals and to prevent degeneration from oxidants. Plants of the Rosaceae family are rich in phytochemicals with a promising chemical structure for various biological activities, including anti-radical scavenging and antioxidant effects [17,20,25,110,111,112,113,114,115]. Three main types of assays are widely used in the study of the antioxidant activities of rose extracts: (i) tests for electron and/or radical scavenging and metal ion chelating (most in vitro); (ii) tests for the inhibition of lipid peroxidation (LP) (in vivo and in vitro); and (iii) tests of the influence of antioxidant enzyme activities (in vivo and in vitro) (Table 4).

Essential oils from Bulgarian *R. damascena* Mill. and *R. alba* L. [21,79], oil, aqueous, alcoholic, and hydroalcoholic extracts [27,110,115], as well as methanol/water extract of fresh petals or flowers from *R. damascena* from Turkey [27,115], Iran [71], Egypt [115,117], and India [25], showed a dose-dependent anti-radical effect against 2,2-diphenyl-1-picrylhydrazyl (DPPH) radicals, significantly higher than that of standard vitamin E, BHT, and BHA [21,71,110]. In vivo tested antioxidant properties of aqueous extracts from rose flowers demonstrated ferric-reducing antioxidant power (FRAP) ability close to that of the control group [71]. It turns out that the antioxidant potential depends on the type of organic extract. Among the fractions obtained by extraction with organic solvents (petroleum ether, chloroform, acetone, methanol, and water) of lyophilized powder from the fresh juice of *R. damascena* Mill. flowers, the acetone fraction is the most active: it inhibits 50% of the superoxide anion radical (•O_2_^−^) production, hydroxyl radical (•OH) generation, and LP, in a concentration-dependent manner [25].

The total phenolic content (TPC) of the methanol extracts of fresh flowers (FFs) and spent flowers (SFs) of the *R. damascena* grown in Turkey is totally different. The FF extract has a TPC five times higher than that of the SF. The antioxidant activity (AOA), tested by a method based on the formation of a phosphomolybdenum complex and the ability for inhibition of DPPH radicals of the FF extract, is higher than that of SF extract, and a positive correlation with the TPC is observed [27]. A semi-solid ethanol extract of dried flowers of *R. centifolia* from India demonstrated a dose-dependent acceptance of DPPH, good AOA, as well as the potential to scavenge superoxide anion radicals (•O_2_^−^) close to that of standard ascorbic acid, as well as marked ferrous ions Fe^2+^chelating activity [118]. Undoubtedly, the DPPH radical is absent in biological systems, but this approach is indicative of the content of phytochemicals in the rose extracts that are able to donate a hydrogen atom, and thus neutralize free radicals [122,123,124,125,126,127].

There are data showing that the composition of essential oils from the *R. damascena* Mill. grown in different geographical areas (Bulgaria, Turkey, Morocco, Moldova, Iran, and China) differs in qualitative analysis and depends on epigenetic factors and the production technology [20,93]. Despite the differences in the composition of the rose oils, its main components are geraniol, citronellol, and nerol. According to Ruberto and Baratta (2000), these dominant components are the reason for the similarity in the biological activities of essential oils [128]. Thus, the differences in the antioxidant properties of oils from *R. damascena* Mill. grown in different areas could be due to the different ratios of the main components, geraniol, citronellol, and nerol, in their composition [117].

UV radiation is a very harmful exogenous factor that induces ROS formation, and treatment with antioxidants is reasonable. Karamalakova et al. (2018) studied antiradical and antioxidant properties before and after UV and γ- irradiation of rose oil using electron donation potential estimation assays, DPPH, ABTS, and Nitric oxide ion scavenging assays, as well as assessing the protection of lipid membranes against radiation damage. Samples after irradiation showed a significant reduction in their donor potential in comparison to non-irradiated oil; nevertheless, in all studied concentrations the ability for scavenging DPPH is higher than the positive control of quercetin. Both γ-irradiated and non-irradiated oil samples demonstrated an increase in scavenging abilities towards ABTS and NO in a concentration-dependent manner [112,121].

*R. damascena* Mill. oil exhibits good antioxidant effects, not only in chemical and biological model systems but also in experimental models of oxidative stress in mice. In *ex vivo* experiments with brain homogenates of mice treated withL-3,4-dihydroxyphenylalanine (L-DOPA), which causes increasing protein oxidation and LP, it was observed that the markers of oxidative stress were reduced upon pretreatment of the experimental animals with antioxidants vitamin C, Trolox, and *R. damascena* Mill. essential oil. The effect of the combination of L-DOPA with damask rose oil was very similar to that of vitamin C and Trolox [114]. Nazıroğlu et al. (2013) reported that vapor of oil from *R. damascena* Mill. lowers the induced chronic mild stress (CMS) LP in rat cerebral cortices and restores concentrations of vitamin A, vitamin E, vitamin C, and b-carotene after depression. The vapor of *R. damascena* oil showed the same effect as citronellol, geraniol, and nerol [116].

### 4.6. Antineoplastic Activities

Regarding the four *Rosa* species upon which this review is focused, research on the antineoplastic activity has been done only for *R. damascena* (and its variety the Taif rose) and *R. gallica* (more precisely its draft in *R. canina*). Still, these studies amount to more than a dozen (see Table 5). Rose essential oils pass easily through and permeabilize the cell membrane and coagulate the cytoplasm, thus damaging lipids and proteins [75,129]. Essential oils stimulate depolarization in mitochondrial membranes and also change their fluidity [130], thus leading to necrosis and apoptosis [24]. The main compounds in rose essential oil are citronellol, linalool, geraniol, flavonoids, and citral, which have been reported to have anticancer activities together with monoterpenes and sesquiterpenes [131,132,133,134]. Terpenes in essential oils could change the nature of the cell membrane, thus causing cell death [24].

MTT assay is the most frequently used test for assessing inhibitory concentration 50 (IC_50_) on transformed cell cultures and the treatment incubation time can be different (12, 48, or 72 h). It was used for examining *R. damascena* Mill. [135] from Iran (e.g., the city of Kashan) [24], Turkey [136], Shaanxi province in China [137], India [138], Bulgaria [139], etc. In addition, micronucleus assay showed that the essential oil from the roses from Kashan had significant cytotoxic and genotoxic effects in peripheral blood lymphocytes, only at doses 50–200 μg/mL [24].

There are several studies of the anticancer activities of *R. damascena* from Iran and other countries. Sleep disorders are among the most common medical complaints in cancer patients. A randomized, single blind-controlled clinical trial was carried out, in which patients smelled essential oil for 20 min. Sleep quality was significantly improved with 5% and 10% oil in comparison to the control group with regard to sleep latency and duration. This effect could be due to the previously reported anti-anxiety and anti-stress properties of rose oils [140]. Self-nanoemulsifying drug delivery systems (SNEDDSs) loaded with essential oil were about three times more cytotoxic, compared to the pure oil, and were preliminary attributed to enhancements in the solubility of the extracts. Apoptosis was also enhanced as shown by a TUNEL assay. SNEDDSs were prepared from a mixture of castor oil, tween 80, polyethylene glycol, and water [135]. Alpha 1,2-mannosidase is a key enzyme in N-glycan processing in the endoplasmic reticulum and Golgi apparatus, and has been one of the enzymes targeted in the development of anticancer therapies. An enzyme assay showed that methanol and aqueous extracts from floret exhibited a 57% noncompetitive inhibition of the enzyme and the aqueous extract inhibited it by 26% [141].

Ethanol extract (70%) of Armenian rose petals was separated into two coumarin and three phenol glycoside fractions and together with the water extract they were examined by a trypan blue exclusion test. Two of the three phenol glycoside fractions significantly inhibited mice Ehrlich ascites carcinoma (EAC) cells by approximately 95% and 85%, respectively, while the other fractions were inhibited by approximately 50–70% [142] *R. damascena* 50% hydroglycol extract induced apoptosis in the epidermoid carcinoma cell line A431 at 10 μg/μL, as established by Hoechst fluorescent staining dye [143]. It is important to note that the vapor phase of the essential oil was cytotoxic to gastric cancer cell line MKN45, to colon cancer cell line SW742, and normal human fibroblast cells, and even the viability of the inner control (untreated wells in the plate) decreased to less than 10%. Apoptosis was the main mechanism of cytotoxicity. On the contrary, the water-soluble phase of the oil increased the cell viability of all cell lines. Considering the inner control as 100%, doses from 1 μL to as high as 60 μL increased the viability of MKN45 in a dose-dependent manner to about 150%. Additionally, when 10 μL was added to the other cell lines the viability increased by about nine-fold. Lower extract volumes (2 or 3 μL) acted as a potent growth factor for the normal fibroblast cells. All of this was revealed by microscopic studies with inverted microscope, MTT assay and flow cytometry [144,145].

A variety of *R. damascena*, *R. damascena* Mill.var. *trigintipetala* Dieck (Taif rose) from the city of Taif, Saudi Arabia, has been studied and essential, concrete, and absolute rose oils have exerted anticancer activity against HepG2 and MCF7 and were cytotoxically and genotoxically safe at a dose of 10 μg/mL on normal human blood lymphocytes. The methods used involved a sulphorhodamine-B (SRB) assay (for the tumor cell lines) and 7-AAD viability assay, DNA index determination. Genotoxicity/antimutagenicity assays included using mitomycin C as a mutagenic positive control, colcemid for obtaining metaphases, and microscopical screening for chromosomal aberrations for determining the mitotic index (for the lymphocytes). Four different chemical categories, namely monoterpenes (68%), sesquiterpenes, and aromatic and aliphatic hydrocarbons were found. The major compounds were β-citronellol (17.6%), geraniol (11.4%), nonadecane (6.5%), nerol (6.4%), linalool (5.9%), α-pinene (4.5%), and phenylethyl alcohol (3.6%) High level of phenyl ethanol may be one of the constituents responsible for the anticancer properties, along with the previously identified geraniol and eugenol [75,146]. Crude methanol extract from fresh and dried roses, and its *n*-butanol and aqueous fractions from the Taif, rose were tested once again by SRB assay with 48 h incubation toward HepG2 cells. No correlation appeared between the anticancer activity and the total phenolic, flavonoid, and flavonol contents [147].

*R. damascene* “Alexandria” and the *R. gallica* “Francesa” draft in *R. canina* were studied through SRB assay with a 48 h incubation, and were found to be moderately cytotoxic against tumor cell lines, while no hepatoxicity towards liver primary culture was observed. Flavonoids were the predominant compounds [148]. In conclusion, it is difficult to identify the reports with the most potent antineoplastic oils, fractions, or extracts, because of the different methods, expression of activity (percentage of survival, IC_50_, etc.), measuring units (μg/mL, μM, etc.), and incubation times applied in the different articles. Still, we have tried to perform an analysis of the information presented in Table 5. Regarding IC_50_, an isoprenylated aurone (IC_50_ on acute myeloid leukemia NB4 cells 4.8 μM and on neuroblastoma SHSY5Y cells 3.4 μM) [137] from *R. damascena* and crude methanol extract and its aqueous fraction from Taif rose (IC50 of 9 and 8 µg/mL on hepatocarcinoma cells, respectively) [147] were the most active. All components that have IC_50_ values lower than 20 µg/mL are also very prospective anticancer candidates according to the guidelines of the National Cancer Institute. The components from the same species that decreases cancer cell viability to less than 15% were all from *R. damascena* a vapor phase of the essential oil applied on gastric (MKN45) and colon (SW742) cancer cell lines [144,145]; essential oil extract (2%) loaded in a self-nanoemulsifying drug delivery system and applied on breast MCF7 and pancreatic PANC1 cancer cell lines [135]; and phenol glycoside fractions of ethanol extract applied on mice Ehrlich ascites carcinoma (EAC) cells [142].

## 5. Conclusions

The present review of the scientific literature provides a systematic summary of the chemical composition and different pharmacological activities of four old rose species: *R. damascena* Mill., *R. alba* L., *R. centifolia* L., and *R. gallica* L. Despite their common ancestors, the four species differ in color, the morphology of colors, and the content of their essential oils and extracts, as well as their antimicrobial, antiviral, and antioxidant effects. What they have in common is that they all are important raw materials for producing cytotoxically and genotoxically safe commercial products. The analysis of the scientific literature in this review shows that essential oils, hydrolates, and extracts of these four rose species, as well as their main compounds, have a promising biological potential to act as natural healers. Natural products have always played a pivotal role in new drug discovery. Due to their valuable therapeutic properties, the studied old rose species are able to act as symptomatic drugs in the prophylaxis and treatment of many diseases. A great challenge for future research in pharmacology and medical fields could be the study of roses from farms in preclinical and clinical trials.

## Figures and Tables

**Figure 1 biomolecules-11-00127-f001:**
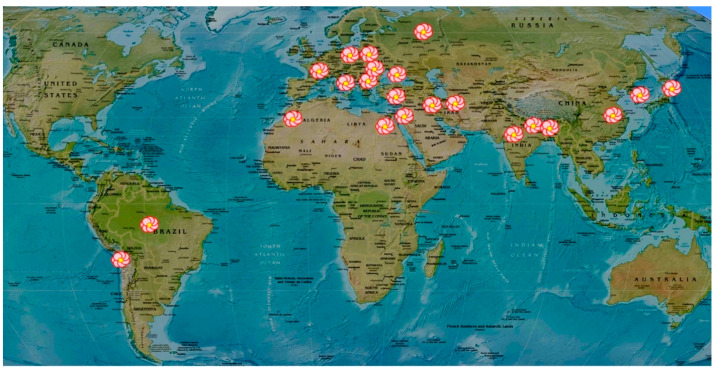
Main locations of rose habitats and producers of rose oils and their products worldwide.

**Figure 2 biomolecules-11-00127-f002:**
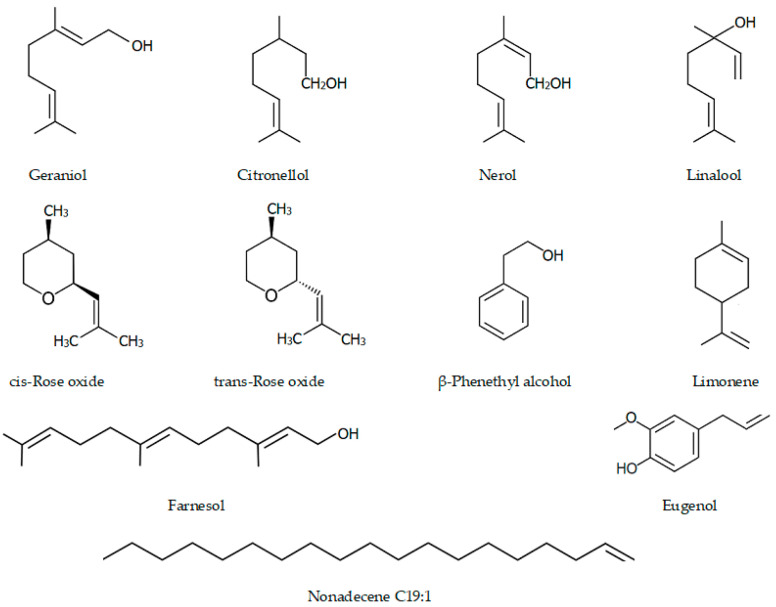
Main compounds in the chemical composition of rose extracts.

**Table 1 biomolecules-11-00127-t001:** The main oil-bearing roses used for rose oil production and the biological activities of their major compounds.

№	Component	Roses and the biological activities of their major compounds							
		*R. damasacena* Mill.	*Rosa alba* L.	*Rosa gallica* L.	*Rosa centifolia* L.	ISO 9842:2003	Biological Activities		
1	Ethanol	0.80	0.15	0.01	0.00	≤2.0%			
2	Limonene	0.00	0.03	0.16	0.05		Anti-inflammatory activity: inhibits pro-inflammatory mediators present in the inflammatory peritoneal exudate in zymosan-induced peritonitis; decreases ROS production, NF-κB activity, and eosinophil migration; has an antidiabetic effect on hyperlipidemia in mice		[42,43,44]
3	Linalool	1.90	1.43	1.58	1.03		Anticonvulsant, antiepileptic activity		[45]
4	β-Phenethyl alcohol	0.40	0.25	0.36	0.09	≤3.5%	Wide spectrum antimicrobial: effective preservative agent for cosmetics; anti-anxiety-like effects		[46,47]
5	Cis-Rose oxide	0.22	0.08	0.05	0.07		Anti-inflammatory properties: decreased the paw edema induced by carrageenan by the suppression of IL-1β production and leukocyte migration		[48]
6	Trans-Rose oxide	0.13	0.06	0.00	0.04				
7	Citronellol	28.72	16.65	9.22	9.22	20.0–34.0%	Antispasmodic, anti-inflammatory, antibacterial, and antifungal: inhibits the mycelial growth, conidia germination, and fungal growth on nail fragments; active against *C. jejuni*, L. monocytogenes, and *S. enterica*		[49,50,51,52,53]
8	Nerol	4.80	10.10	4.72	4.36	0–12.0%	Systolic pressure falls and decreased heart rate		[54]
9	Geraniol	21.40	30.98	24.84	17.60	15.0–22.0%	Insecticidal, repellent, acaricidal activity, antibacterial, and antifungal		[55,56,57,58]
10	Eugenol	1.00	1.12	0.05	0.74		Antiseptic, antibacterial effect against bacteria, pathogens, and harmful microorganisms		[59,60]
11	Methyl eugenol	1.30	0.91	0.94	0.56		Anesthetic in rodents; insect attractant		[61]
12	n-Heptadecane	1.40	2.00	2.98	1.07	1.0–2.5%			
13	Farnesol	1.90	2.98	1.26	3.48		Anti-inflammatory; anticancer properties		[62]
14	Nonadecene C_19:1_	2.30	4.35	1.25	2.28				
15	n-Nonadecane C_19_	11.10	12.14	22.67	8.10	8.0–15.0%			
16	n-Eicosane C_20_	1.00	1.01	1.06	0.55				
17	n-Heneicosane C_21_	5.00	10.21	9.07	6.31	3.0–5.5%			
18	n-Tricosane C_23_	1.30	0.81	2.58	5.90				
19	n-Pentacosane C_25_	0.50	0.23	1.04	2.86				
20	n-Heptacosane C_27_	0.40	0.10	0.55	1.79				

**Table 2 biomolecules-11-00127-t002:** Cytotoxic/anticytotoxic, mutagenic/antimutagenic, and genotoxic/anti-genotoxic potential of extracts and oils of the studied *Rosa* species.

*Rosa* Species	Type of Extract	Test Methods	Test Objects	Biological Activity						References
				T/CT	GT	MG	ATC	AGT	AMG	
*R. alba* L. (Bulgaria)	Essential oil	Chromosome aberration assay,	*H. vulgare* root tip meristems	−	−		+	+		[77,78]
		Micronucleus assay,					+			
		Comet assay	Human lymphocytes					+		
*R. damascena* white variety (Iran)	Aqueous and methanol extracts	Modified MTT assay,	HeLa cells line NCBI,	+						[71]
			Human lymphocytes,	−						
		Hematology and clinical chemistry parameter analysis	Wistar rats	−						
*R. damascena* Mill. (Iran)	Essential oil	MTT assay,	Normal NIH3T3,	+	+					[24]
			Cancer cells A549,	+	+					
		Micronucleus assay	Peripheral blood lymphocytes	+	+					
*R. damascena* Mill. (Iran)	Ethanol extract	Modified MTT assay	HeLa cell line	+						[73]
*R. damascena* Mill.	Aqueous and ethanol extract	MTT assay, BrdUassay, TUNEL assay	Human gastric cancer cell line AGS cells	+						[72]
*R. damascena trigintipetala* Dieck (Saudi Arabia)	Concrete and absolute rose oil	Viability assay,	Human lymphocytes	−	−				+	[75]
		Chromosome aberration assay,								
		Sulphorhodamine-B (SRB) assay	Cell lines HepG2 and MCF7	+						
*R. damascena* and *R. centifolia * (Bulgaria)	Aqueous-alcoholic extracts	Double-staining fluorescence assay	Human lymphocytes	−	−			+		[74]
*R. damascena* Mill.	Essential oil	MTT assay	A549, PC-3, MCF-7 human tumor cell lines	+						[76]
*R. centifolia* cultivars, “passion,” “pink noblesse”, and “sphinx”	Aqueous extract, Rose tea from petals	*E. coli* RNA polymerase B (rpoB)-based Rif S→Rif R (rifampicin sensitive to resistant) forward mutation assay	*E. coli*						+	[79]

T/CT: toxicity/cytotoxicity; GT: genotoxicity; MG: mutagenicity; ATC: anticytotoxicity; AGT: anti-genotoxicity; AMG: antimutagenicity. All statements about the effectiveness of the examined markers are concentration-dependent and relate to the specific parameters of the arrangement of the test. Note: the cytotoxic effects against cancer cells are given in the count of T/CT.

**Table 3 biomolecules-11-00127-t003:** Antimicrobial activity of rose essential oils and extracts.

*Rosa* Species	Type of Extract	Test Methods	Observations (DIZ in mm, MIC and MBC in µg/mL, respectivley mg/mL^−1^ or in %)	References
*R. damascena* Mill. (Turkey)	10% MeOH extracts from fresh flower^1^ and 10% MeOH extracts from spent flower^2^	Agar diffusion method	*A. hydrophila* (^2^18 mm), *B. cereus* (^1^16 mm), *E. feacalis* (^1^15 mm), *E. coli* (^1^17 mm, ^2^16 mm), *M. smegmatis* (^1^15 mm, ^2^21 mm), *P. vulgaris* (^1^18 mm, ^2^15 mm), *P. aeruginosa* (^1^16 mm), *P. fluorescens* (^2^15 mm), *S. enteritidis* (^1^21 mm, ^2^16 mm), *S. typhimurium* (^1^17 mm), *S. aureus* (15 mm), *Y. enterocolitica* (^1^15 mm)	[27]
*R. damascena* Mill. (Turkey)	5% MeOH extracts from fresh flower^1^ and 5% MeOH extracts from spent flower^2^	Agar diffusion method	*A. hydrophila* (^2^15 mm), *E. coli* DM (^1^16 mm), *M. smegmatis* (^2^18 mm)	[27]
*R. damascena* Mill. (Bulgaria)	Essential oil	Disc diffusion and serial broth dilution methods	*B. cereus* (24 mm, MIC = MBC = 128 µg/mL), *S. aureus* (18 mm, MIC = MBC = 256 µg/mL), *S. epidermidis* (21 mm, MIC = MBC = 256 µg/mL), *E. coli* (17 mm, MIC = 512 µg/mL, MBC = 1024 µg/mL), *C. albicans* (17 mm, MIC = 1024 µg/mL, MBC = 2048 µg/mL), *C. albicans* (15 mm, MIC = 1024 µg/mL, MBC = 2048 µg/mL)	[93]
*R. damascena* Mill. (Turkey)	Essential oil	Disc diffusion method	*B. cereus* (22 mm), *S. aureus* (16 mm), *S. epidermidis* (20 mm), *E. coli* (16 mm), *C. albicans* (16 mm)	[93]
*R. damascena* Mill. (Morocco)	Essential oil	Disc diffusion method	*B. cereus* (20 mm), *S. epidermidis* (18 mm)	[93]
*R. damascena* Mill. (China)	Essential oil	Disc diffusion method	*B. cereus* (18 mm), *S. epidermidis* (17 mm)	[93]
*R. damascena* Mill. (Iran)	Essential oil	Serial broth dilution method	MIC = MBC = 1 µg/mL for *S. aureus*,*S. agalactiae*, *S. sanguis*, *S. salivarius*, *E. coli*, *E. aerogenes*, *S. marcescens*; MIC = 0.5 µg/mL and MBC = 1 µg/mL for *S. saprophyticus*, *S. epidermidis*, *B. cereus*, *B. subtilis*, *S. dysenteriae*, *Shigella flexneri*, *C. albicans*; MIC = 0.25 µg/mL and MBC = 0.5 µg/mL for *S. pyogenes*, MIC = 1 µg/mL and MBC = 2 µg/mL for *E. faecalis*, *E. faecium*, *S. typhimurium*, *P. aeruginosa*; MIC = 0.125 µg/mL and MBC = 0.25 µg/mL for *K. pneumonia*, *P. vulgaris*; MIC = 0.125 µg/mL and MBC = 1 µg/mL for *A. flavus;* MIC = MBC = 0.25 µg/mL for *A. niger*; MIC = 0.5 µg/mL and MBC = 2 µg/mL for *A. parasiticus*	[24]
*R. alba* L. (Bulgaria)	Essential oil	Disc diffusion and serial broth dilution methods	*A. actinomycetemcomitans* (MIC = 0.45 mg/mL^−1^), *S. mutans* (MIC = 0.82 mg/mL^−1^), *E. faecalis* (MIC = 1.36 mg/mL^−1^)	[21]
*R. alba* L. (Bulgaria)	Essential oil	Disc diffusion and serial broth dilution methods	*B. cereus* (14–15 mm, MIC = MBC = 0.05%), *E. coli* (MIC = MBC = 0.2%), *S. aureus* MIC = 0.41%, MBC = 0.82%), *S. epidermidis* (MIC = MBC = 0.1%), *S. abony* (MIC = MBC = 0.82%), *C. albicans* (MIC = MBC = 0.82%), *C. tropicalis* (MIC = 0.41%, MBC = 0.82%)	[8]
*R. centifolia* L. (Morocco)	90% EtOH extract	Disc diffusion and serial broth dilution methods	*L. monocytogenes* ATCC 19117 (20 mm, MIC = 0.9 mg/mL)	[94]
*R. centifolia* L. (Palestine)	Crude EtOH extract	Disc diffusion method	*P. acnes* (19 mm), *E. coli* (31 mm),	[95]
*R. centifolia* L. (India)	EtOH extract	Agar diffusion method	*S. mutans* (15 mm), *C. albicans* (16 mm)	[96]

MeOH: methanol; EtOH: ethanol; DIZ: diameter of inhibition zone; MIC: minimal inhibitory concentrations; MBC: minimal bactericidal concentrations. On this Table the antimicrobial activities ranging from 14 mm and upare presented.

**Table 4 biomolecules-11-00127-t004:** Free radical scavenger and antioxidant activities of rose extracts.

*Rosa* Species	Type of Extract	Observations	References
*R. damascena* Mill. from Gulbirlik Inc. (Isparta, Turkey)	Rose oil vapor of essential oil, obtained by water steam distillation	Inhibits lipid peroxidation (LP) induced by chronic mild stress (CMS) in the rat cerebral cortex homogenate; restores content of vitamin A, vitamin E, vitamin C, and b-carotene in brain cortex homogenate	[116]
*R. damascena* Mill. from Turkey	Essential oil by water steam distillation; Aromatic water (hydrosol) by water steam distillation; Methanol extract of fresh flowers and spent flowers	2,2-diphenyl-1-picrylhydrazyl (DPPH) radical scavenging activity; Fe^2+^metal-chelation activity; ferric-reducing antioxidant power DPPH radical scavenging activity; AOA by the formation of phosphomolybdenum complex	[27,115]
*R. damascena* Mill. from rose gardens of Kashan, city of Iran	Aqueous/methanol extract	DPPH radical scavenging activity; β-carotene bleaching effect; ferric-reducing antioxidant power	[71]
*R. damascena* Mill. from Iran, population of Guilan	Essential oil hydro-alcoholic extract	DPPH radical scavenging activity; LP inhibitory effects	[110]
*R. damascena* Mill. from Bulgaria, Turkey, and Egypt	Essential oil by water steam distillation	DPPH radical scavenging activity	[117]
*R. centifolia* L.from a local market near Jamia Hamdard	Ethanol (70%) extract of dried flowers	Dose-dependent acceptations of DPPH activity; antioxidant activity (AOA) by the formation of phosphomolybdenumcomplex; dose-dependent manner for scavenging superoxide anion radical ferrous iron chelating activity	[118]
*R. gallica* L. (Hungary)	Methanol extracts from dried fruit rosehips Water/ethanol 80/20 *v/v*, 20 °C Aqueous extract	Ferric-reducing antioxidant power (FRAP) activity in blood plasma; the activities are as follow: *R. spinosissima* >*R. canina* >*R. rugosa* >*R. gallica* Ethanol extracts of rosehips have higher phenolic content and antioxidant activity than water extracts	[119]
*R. alba* L. oil, *R. damascena* Mill. oil, and *R. damascena* Mill. oil from rose water (Bulgaria); *R. gallica* L. (Moldova)	Essential oil by water steam distillation	Superoxide anion radicals scavenging activity	[20]
*R. alba* L., *R. damascena* Mill. (Bulgaria)	Essential oil by water steam distillation Hydrosols by water steam distillation	DPPH radical-scavenging activity; Fe^2+^metal-chelation activity; inhibition of Fe^2+^-induced LP in egg liposomes; inhibition of Fe^2+^/asc. acid-nduced LP in egg liposomes; inhibition of hydroxyl radicals generation	[21,79,113]
*R. damascena* Mill. (Bulgaria)	Essential oil by water steam distillation	Inhybition of lipid peroxidaton in brain homogenate and the blood of mice with an experimental model of oxidative stress	[114]
*R. damascena* Mill. from International Flavors and Fragrances Inc. (New York).	Essential oil by water steam distillation	Inhibition of oxidation of hexanal to hexanoic acid in a dose-related manner; DPPH scavenging in a dose-dependent manner; inhibition of the formation ofmalonaldehyde (MDA) from squalene upon UVirradiation	[120]
*R. damascena* Mill. (Thrissur- Kerala, India).	Liophylisate from fresh juice of rose flower, eluted in silica gel column by petiole ether, chloroform, acetone, methanol, and water.	Acetone fraction: inhibits 50% superoxide radical production, hydroxyl radical generation, and inhibits Fe^2+^asc. acid induced lipid peroxidation in liver homogenate of mice, treated by CCl_4_	[25]
*R. damascena* Mill. (fresh and spent flowers, and green leaves), Isparta, Turkey.	Hot extractions with methanol Cold extractions with methanol	Antiradical activity: DPPH, and FRAP methods	[111]
*R. damascena* Mill., Kazanlak, Bulgaria	Essential oil by water steam distillation	Inhibition of UV and γ- irradiation-induced oxygen/nitrogen free radicals; hydroxyl radical scavenging potential; examination by EPR assay, DPPH, 2,2′-azino-bis-3-ethylbenzthiazoline-6-sulphonic acid (ABTS), Nitric oxide scavenging assay, and protection of liposomal lipids from soy lecithin and cholesterol; γ-irradiated and non-irradiated oils demonstrate an increase in scavenging abilities towards ABTS and NO in a concentration-dependent manner	[112,121]

**Table 5 biomolecules-11-00127-t005:** Antineoplastic activity and toxicological data.

*Rosa* Species	Type of Extract, Fraction, or Compound Investigated	Anticancer and Other Activities Supporting the Anticancer Therapy	References	Toxicological Data/Safety	References
***R. damascena* Mill.**	**Essential oil**	– Lung cancer cell line A549 (IC_50_ = 36.4 µg/mL *);	[24]	– No toxic effects in doses up to 10 μg/mL	[24]
		– Significant improvement of cancer patients’ sleep quality regarding sleep latency and duration (5% and 10% essential oil).	[140]	– Low cytotoxicity towards the normal cell line NIH3T3 (IC_50_ = 42.9 µg/mL)	[24]
		– not cytotoxic at 200 μg/mL against tumor cell lines HepG2, Hep3B (hepatoma), A549, and breast MCF-7 and MDA-MB-231 cell lines	[139]	– Significant cytotoxic and genotoxic effects in peripheral blood lymphocytes at doses over 10 µg/mL, 1% (after 1 h exposure) and 0.1% (after 24 h exposure)	[24,74]
	– Vapour and water soluble phases of the essential oil	− Gastric (MKN45) and colon (SW742) cancer cell line: ○ Vapour phase of the essential oil: decreased cell viability to less than 10%; ○ Water soluble phase of the essential oil (1–60 µL): increased cell viability about 9 folds;	[144,145]	− Vapour phase of the essential oil: cytotoxic towards normal human fibroblasts − Water soluble phase of the essential oil (2–3 μL): potent growth factor for normal fibroblasts	[144,145]
	**Extracts:**				
	– Essential oil extract (2%) loaded in a self-nanoemulsifying drug delivery system	– Breast MCF7 (10% cell survival) and pancreatic PANC1 (15% cell survival) cancer cell lines	[135]	N.d. **	-
	– Methanol extract	– Cervical cancer cell line HeLa (IC_50_ = 265 μg/mL): a selectivity index towards normal kidney epithelial Vero cells SI > 3.8 (SI > 2 means selective toxicity)	[136]	– Low cytotoxicity on normal kidney epithelial Vero cells (IC_50_>1 g/mL)	[136]
	– 50% methanol extract	– Not cytotoxic at 2 μg/mL for 72 h towards U937 human lymphoma cell line.	[138]	N.d.	-
	– 50% ethanol extract of flowers	– Cervical cancer cells HeLa (IC_50_ (72 h) = 305.1 μg/mL, IC_50_ (48 h) = 1540 μg/mL, IC_50_ (24 h) = 2135 μg/mL for 24 h)	[73]	N.d.	[73]
	– 70% ethanol extract of rose petals	– Mice Ehrlich ascites carcinoma (EAC) cells (IC_80_ = 15 μg/mL, 80% growth inhibition at 200 μg/mL);	[142]	– Cytotoxic to mice peripheral blood leukocytes: 90% growth inhibition at 200 μg/mL	[142]
	– 50% hydroglycol extracts	– Epidermoid carcinoma cell line A431 (IC_50_ = 3220 μg/mL)	[143]	N.d.	-
	– Water extract of petals	– No cytotoxicity towards EAC cells	[142]	– Not cytotoxic to peripheral blood leukocytes	[142]
	**Fractions:**				
	– Coumarin fractions of 70% ethanol extract	– Inhibition of EAC cells by approximately 50–70%	[142]	– About 30% inhibition of peripheral blood leukocytes for the monomer fraction	[142]
	– Phenol glycoside fractions of 70% ethanol extract	– Significant inhibition of EAC cells by approximately 85–95% for two of the fractions	[142]	– No significant inhibition of peripheral blood leukocytes	[142]
	**Active substance**				
	– Isoprenylated aurone	– Acute myeloid leukemia NB4 (IC_50_ = 4.8 μM) and neuroblastoma SHSY5Y (IC_50_ = 3.4 μM) cell lines; less cytotoxic to lung A549, prostate PC3 and breast MCF7 cells	[137]	N.d.	-
***R. damascena* Millervar. *trigintipetala* Dieck (Taif rose)**	**Essentialoils, concrete, and absolute**	− Hepatocarcinoma HepG2 cell line: ○ Essential oil: IC_50_ = 13 μg/mL *** ○ Concrete oil: IC_50_ = 16.28 μg /mL *** ○ Absolute oil: IC_50_ = 24.94 μg /mL − Breast cancer MCF7 cell line: ○ Essential oil: IC_50_ = 16 μg/mL *** ○ Concrete oil: IC_50_ = 18.09μg/mL *** ○ Absolute oil: IC_50_ = 19.69 μg /mL ***	[75,146]	– Essential oil was cytotoxically and genotoxically safe at a dose of 10 μg/mL on normal human blood lymphocytes; concrete oil was even less toxic than absolute oil which showed significant antimutagenic activity at a dose of 10 μg/mL	[75]
	**Extracts:**				
	– Crude 80% methanol extract from fresh and dried roses	– HepG2 cells (fresh rose extract IC_50_ = 9 μg/mL ***, dry rose extract IC_50_ = 13 μg/mL ***)	[147]	– N.d.	-
	**Fractions of crude 80% methanol extract from fresh and dried roses:**				
	– *n*-butanol	– HepG2 cells (fresh rose extract IC_50_ = 17 μg/mL ***, dry rose extract IC_50_ = 18 μg/mL ***)	[147]	– N.d.	-
	– aqueous	– HepG2 cells (fresh rose extract IC_50_ = 8 μg/mL ***, dry rose extract IC_50_ = 11 μg/mL ***)			
***R. damascena*** **“Alexandria” and** ***R. gallica* “Francesa” draft in *R. canina***	**Methanol: water (80:20) extract** (from petals)	– HeLa (IC_50_ = 308 μg/mL) and HepG2 (IC_50_ = 297 μg/mL) cellslarge cell lung carcinoma cell line NCI-H460 (IC_50_> 400 μg/mL)	[148]	– No hepatoxicity towards the non-tumor porcine liver primary culture (IC_50_ > 400 μg/mL).	[148]
	**Infusion** (from petals)	– MCF-7 (IC_50_ = 377 μg/mL) and HepG2 (IC_50_ = 315 μg/mL) cellsNCI-H460 cells (IC_50_> 400 μg/mL)	[148]	– N.d.	-

* All IC_50_ values presented in the table are obtained using the MTT dye assay or sulphorhodamine-B assay; ** N.d.: no published data; *** Values fall within the National Cancer Institute (NCI) guidelines for a promising agent (IC_50_ < 20 μg/mL).

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
