# Peer review of "Rose Flowers—A Delicate Perfume or a Natural Healer?"

_biomolecules, 2021, doi:10.3390/biom11010127_

Round 1
Reviewer 1 Report
8 December -2020
Journal: Biomolecules
Title: Rose flowers - a delicate perfume or a natural healer?
Authors: Milka Mileva, Yana Ilieva, Gabrilele Jovtschev, Svetla Gateva, Maya Zaharieva, Almira Dimitrova, Ludmila Dimitrova, Ana Dobreva, Tsveta Angelova, Nelly Vilhelmova-Ilieva,Violeta Valcheva and Christo Najdenski
The authors have discussed the phytochemical profile of four roses species, Rosa damascena Mill., R. alba L., R. centifolia L., and R. gallica L. In addition, their habitats, and biological activities have been investigated. The manuscript could be accepted after major revision.
Comments to Authors:
- The graphical abstract is unclear and its written text is not representative. Please improve the quality of the graphic abstract.
- The authors would add the originality of the work.
- The abstract should be rewritten in a more precise and comprehensive way.
- Please add a brief conclusion to the end of the abstract.
- What do you mean by technological products?
- " Rosa damascena, Rosa alba L., Rosa centifolia L., and Rosa gallica L.; the authors should use the full name of the genus for the first time after that they could use abbreviation i.e., R. alba L. The same applies for all the other abbreviations throughout the text, you could also add a list of abbreviations if needed.
- Keywords should be more specific.
- Line 31: " in the 13th century BC" should be " in the 13th century BC".
- “as early as the time of Ramses The Second” could be “as early as the time of Ramses II”.
- Line 33: “the most beautiful of the goddesses”, do you mean “the most beautiful goddesses”?
- Lines 34 -35, who called the roses these nicknames?
- Line 37: What are evergreen lianas? please cite a proper reference.
- The authors would expand on the rose’s technological products part.
- Lines 54 and 58: The authors would focus on the four species that are targeted in the abstract and avoid mention other species except if it adds up to the content and related to the topic. For instance, “Rosa rugosa” and “Rosa chinensis”.
- Line 59, if you will use the term “In recent years”, you have to make sure that all the references were from the immediate past.
- The authors could explain briefly the origin, morphology, traditional uses, chemical constituents, biological activities, and applications of the different four roses species.
- " A short on habitat”, please check the syntax.
- Lines 74-75: “Most of its species are native to Asia”, why did you mention species as plural where you are talking about only singular species?
- Please, be specific regarding the habitat and cultivation of damascene as you are talking in the whole section in general.
- Line 85: The authors would make sure that their writing is from the oldest to the newest.
- Your writing from line 102 till 109 is the best, you can apply this style to the related sections.
- From line 116 till 121, you shouldn’t talk about chemical constituents, pharmacological properties, and the applications of this species and instead explain briefly regarding the habitat and cultivation of the species.
- Table 1, the authors would add an extra column for the compounds’ biological activity.
- Table 2, It is better to focus only on the biological activity of the four roses species.
- You don’t have to mention the Rosa canina cytotoxic and anti-proliferative activities in lines 188 and 195.
- Lines 169- 249, should be rewritten in a more organized way and better to mention the activity related to each species in a separate section.
- Line 213: " Essential oil of Bulgarian Rosa alba L. (250, 500 and 1000 mg/mL)" are of very high concentration.
- A lot of data for biological activity demonstrates high concentration; high concentration is an indication of weak activity
- The mechanism of action for each assay would be mention.
- The best extraction way from each species that showed the best activity should be mentioned and the authors' conclusion at the end of each activity is highly recommended.
- Lines 252- 255, don't represent a relevant introduction and could be removed.
- The observation column in table 3, should be summarized focusing only on the highest activity.
- The authors could improve their writing style. For example, "As can see" in line 436.
- The whole name of the FRAP assay?
- Line 488, The aqueous and methanol extracts, please cite a reference.
- All the biological activities should be more focused on the four roses species, and the same applies to the tables.
- More figures are highly recommended. Improve the quality of figure 1.
- Table 1 is unclear; what are the numbers and ISO stand for?
- Table 5, is very crowded and un
- The authors include a delicate perfume in the title but it wasn’t included as one of the study scopes in the manuscript.
- The authors could get benefit from the following references:
- El-Seedi, Hesham R., et al. "Exploring Natural Product-Based Cancer Therapeutics Derived from Egyptian Flora." Journal of Ethnopharmacology (2020): 113626.
- Khalifa, Shaden AM, et al. "Screening for natural and derived bio-active compounds in preclinical and clinical studies: one of the frontlines of fighting the coronaviruses pandemic." Phytomedicine (2020): 153311.
- Akram, Muhammad, et al. "Chemical constituents, experimental and clinical pharmacology of Rosa damascena: a literature review." Journal of Pharmacy and Pharmacology 72.2 (2020): 161-174.
- In general:
- Please follow the journal instruction of the authors.
- The review has a lot of data and unfocused, should be more organized.
- The review lacks originality; it illustrates a collection of data.
- A comprehensive critical analysis is missing.
- Many parts without references.
- The authors should use a list of abbreviations.
- The authors would unify the style of the references based on the journal instructions.
- Please check reference number 27. In reference number 29, the journal name is missing.
- Most of the cited references journals are very weak.
- English editing is highly required.
Author Response
We are thankful to the reviewers for the comprehensive and critical review, which will increase immeasurably the value of the manuscript. According to the suggestions of the reviewers we made the following changes:
Comments to Authors: Reviewer 1
- The graphical abstract is unclear and its written text is not representative. Please improve the quality of the graphic abstract. This is done.
- The authors would add the originality of the work. This is done.
- The abstract should be rewritten in a more precise and comprehensive way. This is done.
- Please add a brief conclusion to the end of the abstract. This is done.
- What do you mean by technological products?
- " Rosa damascena, Rosa alba L., Rosa centifolia L., and Rosa gallica L.; the authors should use the full name of the genus for the first time after that they could use abbreviation i.e., R. alba L. The same applies for all the other abbreviations throughout the text, you could also add a list of abbreviations if needed. This is done.
- Keywords should be more specific. This is done.
- Line 31: " in the 13th century BC" should be " in the 13th century BC". This is corrected.
- “as early as the time of Ramses The Second” could be “as early as the time of Ramses II”. This is corrected.
- Line 33: “the most beautiful of the goddesses”, do you mean “the most beautiful goddesses”? This is corrected.
- Lines 34 -35, who called the roses these nicknames?
- Line 37: What are evergreen lianas? please cite a proper reference. The article has been added - the source is in Russian.
- The authors would expand on the rose’s technological products part. This is done.
- Lines 54 and 58: The authors would focus on the four species that are targeted in the abstract and avoid mention other species except if it adds up to the content and related to the topic. For instance, “Rosa rugosa” and “Rosa chinensis”. This is done.
- Line 59, if you will use the term “In recent years”, you have to make sure that all the references were from the immediate past. This is corrected.
- The authors could explain briefly the origin, morphology, traditional uses, chemical constituents, biological activities, and applications of the different four roses species. This is done.
- " A short on habitat”, please check the syntax. This is done.
- Lines 74-75: “Most of its species are native to Asia”, why did you mention species as plural where you are talking about only singular species? This is corrected.
- Please, be specific regarding the habitat and cultivation of damascene as you are talking in the whole section in general. This is corrected.
- Line 85: The authors would make sure that their writing is from the oldest to the newest. This is corrected.
- Your writing from line 102 till 109 is the best, you can apply this style to the related sections. This is done.
- From line 116 till 121, you shouldn’t talk about chemical constituents, pharmacological properties, and the applications of this species and instead explain briefly regarding the habitat and cultivation of the species. This is done.
- Table 1, the authors would add an extra column for the compounds’ biological activity. This is done.
- Table 2, It is better to focus only on the biological activity of the four roses species. – This is done.
- You don’t have to mention the Rosa canina cytotoxic and anti-proliferative activities in lines 188 and 195. This is removed.
- Lines 169- 249, should be rewritten in a more organized way and better to mention the activity related to each species in a separate section. This is corrected.
- Line 213: " Essential oil of Bulgarian Rosa alba L. (250, 500 and 1000 mg/mL)" are of very high concentration. – The concentrations are corrected - " Essential oil of Bulgarian Rosa alba L. (250, 500 and 1000 μg/mL)".
- A lot of data for biological activity demonstrates high concentration; high concentration is an indication of weak activity. - The most effective concentration of the rose extracts depend on the type of extract or essential oil, as well as the area of application and the cell sensitivity. Most of the extracts and oils possess high activity in cancer cells where LC50 is lower than that in normal cells. These results indicate that the rose extracts and oils are safe or low toxic to normal cells and could be successfully apply in the cancer therapy.
- The mechanism of action for each assay would be mention. – In the text we mention that the MTT test, BrdU assay, double-staining fluorescence assay, TUNEL assay give information about the toxic/cytotoxic activity of the extracts and oils. The reverse mutations, somatic mutation and the recombination test (SMART) are applied to study the mutagenic/antimutagenic activity, and to detect genotoxic/antigenotoxic potential, the tests for induction of chromosomal aberrations, micronuclei as well as comet assay, etc. are used. The assays for cytotoxicity assess the cell viability, proliferative activity and induction of apoptosis. The most common assays for genotoxicity show the mutation frequency, changes in the DNA integrity, changes in the chromosome structure, as well as disturbances in the course of mitosis.
- The best extraction way from each species that showed the best activity should be mentioned and the authors' conclusion at the end of each activity is highly recommended. - It is difficult to make general statements about the best extraction variant, as well as the most effective concentration of the essential oils. There are several reasons for this: i) there are not enough examinations in the available literature; ii) The most effective concentrations are object-specific, and iii) depend on the area of application. In this sense, specific studies are necessary for each application.
- Lines 252- 255, don't represent a relevant introduction and could be removed. This is removed.
- The observation column in table 3, should be summarized focusing only on the highest activity. This is done.
- The authors could improve their writing style. For example, "As can see" in line 436. This is done.
- The whole name of the FRAP assay? This is done.
- Line 488, The aqueous and methanol extracts, please cite a reference. This is done.
- All the biological activities should be more focused on the four roses species, and the same applies to the tables. This is done.
- More figures are highly recommended. Improve the quality of figure 1. This is done.
- Table 1 is unclear; what are the numbers and ISO stand for? ISO 9842 is cited.
- Table 5, is very crowded. Table 5 was re-written and includes only the antineoplastic activity with the relevant toxicological data and IC50 values in order to compare on a clearer way the activities of the different extracts, fractions and BAC – the values indicating promising high activity are denoted with *** and an explanation is given in the legend under the table.
- The authors include a delicate perfume in the title but it wasn’t included as one of the study scopes in the manuscript. We have explained that essential oils and extracts are used in cosmetics and perfumery, but they also have a healing effect.
- The authors could get benefit from the following references: The authors express their special gratitude for this recommendation. We used the mentioned articles, they were very valuable for the design of the manuscript.
- El-Seedi, Hesham R., et al. "Exploring Natural Product-Based Cancer Therapeutics Derived from Egyptian Flora." Journal of Ethnopharmacology (2020): 113626.
- Khalifa, Shaden AM, et al. "Screening for natural and derived bio-active compounds in preclinical and clinical studies: one of the frontlines of fighting the coronaviruses pandemic." Phytomedicine (2020): 153311.
- Akram, Muhammad, et al. "Chemical constituents, experimental and clinical pharmacology of Rosa damascena: a literature review." Journal of Pharmacy and Pharmacology 72.2 (2020): 161-174.
- In general:
- Please follow the journal instruction of the authors. This is done.
- The review has a lot of data and unfocused, should be more organized. This is done.
- The review lacks originality; it illustrates a collection of data. This is done.
- A comprehensive critical analysis is missing. This is done.
- Many parts without references. This is corrected.
- The authors should use a list of abbreviations. This is done.
- The authors would unify the style of the references based on the journal instructions. This is done.
- Please check reference number 27. In reference number 29, the journal name is missing. This is done.
- Most of the cited references journals are very weak. This is corrected.
- English editing is highly required. This is done.

Reviewer 2 Report
The provided manuscript with ID: biomolecules-1040788 and entitled "Rose flowers - a delicate perfume or a natural healer?" is simply a good review article which updates and summarizes the phytochemical composition of the volatile oils obtained from flowers of four widely distributed Roses in Bulgaria as well as the flower extracts and their biological activities.
The authors indicated that the volatile oils from these four Roses are widely used as perfumes as well as natural healers.
There is a potential usages of the flowers for human health. Actually, this is a valuable review for natural products chemists, researchers and scientists. A very good effort had been done by the authors.
The only comments:
- The tables with the biological activities need to be rearranged to look better.
- The manuscript is long and needs reduction.
- Several spelling mistakes need to be corrected.
Author Response
We are thankful to the reviewers for the comprehensive and critical review, which will increase immeasurably the value of the manuscript. According to the suggestions of the reviewers we made the following changes:
- The tables with the biological activities need to be rearranged to look better. This is corrected.
- The manuscript is long and needs reduction. This is corrected.
- Several spelling mistakes need to be corrected. This is corrected.
Round 2
Reviewer 1 Report
29 December -2020
Journal: Biomolecules
Title: Rose flowers - a delicate perfume or a natural healer?
Authors: Milka Mileva, Yana Ilieva, Gabrilele Jovtschev, Svetla Gateva, Maya Zaharieva, Almira Dimitrova, Ludmila Dimitrova, Ana Dobreva, Tsveta Angelova, Nelly Vilhelmova-Ilieva,Violeta Valcheva and Christo Najdenski
Comments to Authors:
- English editing is still highly recommended; for instance, “voluble” in lines 20 and 25, respectively.
- "Essential oils and extracts with their valuable therapeutic properties respiratory antiseptics, anti-inflammatories, mucolytics, expectorants, decongestants, antioxidants, are able to act as symptomatic drugs in today's pandemic situation and in this way to alleviate the dramatic sufferings during a Covid19 infection" is this the aim of the review?
- The authors would add the originality of the work.
- What do you mean by technological products?
- " Rosa damascena, Rosa alba L., Rosa centifolia L., and Rosa gallica L.; the authors should use the full name of the genus for the first time it appears in the review then they can use abbreviation i.e., R. alba L.
- Lines 34 -35, who called the roses these nicknames?
- Line 37: what do you mean by evergreen lianas?
- The authors could explain briefly the origin, morphology, traditional uses, and chemical constituents of the four roses species in the introduction part.
- Line 78: "At the beginning of the 19th century" should be "At the beginning of the 19th century".
- The first letter of the words should be capital in all tables’ cells.
- In table 1, unify the style in the biological activity column regarding the mode of action.
- More figures are highly recommended.
- Line 794, “ISO9842: Oil of rose (Rosa x damascena Miller). Available online: (accessed on) please specify.
Author Response
We are thankful to the reviewer for the comprehensive and critical review, which will increase immeasurably the value of the manuscript. According to the suggestions of the reviewers we made the following changes:
- English editing is still highly recommended; for instance, “voluble” in lines 20 and 25, respectively.
A licensed English language editor has reviewed the article.
- "Essential oils and extracts with their valuable therapeutic properties respiratory antiseptics, anti-inflammatories, mucolytics, expectorants, decongestants, antioxidants, are able to act as symptomatic drugs in today's pandemic situation and in this way to alleviate the dramatic sufferings during a Covid19 infection" is this the aim of the review?
The expression has been corrected.
- The authors would add the originality of the work.
The originality of the work was added in all of the chapters.
- What do you mean by technological products?
According to the Law on Rose Production in Bulgaria, technological products are essential oil, hydrolate (hydrosol), absolute and concrete. Obviously, the international scientific community does not understand this term. That is why the authors have replaced it with commercial products.
- " Rosa damascena, Rosa alba L., Rosa centifolia L., and Rosa gallica L.; the authors should use the full name of the genus for the first time it appears in the review then they can use abbreviation i.e., R. alba L. This is done.
- Lines 34 -35, who called the roses these nicknames?
We have indicated the literature where we have read these nicknames.
- Line 37: what do you mean by evergreen lianas?
According to the findings of Nazarenko et al. (1983), the genus Rosa, to which the oil-bearing species belong, originates from the ancient evergreen lianas of Sundarbans in India, called "The pharmacy of the world" because more than 1/4 of the drugs known today in medicine are based on plants from these forests.
- The authors could explain briefly the origin, morphology, traditional uses, and chemical constituents of the four roses species in the introduction part.
This is done. The morphological specifics of the inflorescences are also evident from the graphic abstract, which has been reworked and applied.
- Line 78: "At the beginning of the 19th century" should be "At the beginning of the 19th century". This is done.
- The first letter of the words should be capital in all tables’ cells.
This is done.
- In table 1, unify the style in the biological activity column regarding the mode of action.
This is done.
- More figures are highly recommended.
The formulas of the main components in the extracts are prepared and applied.
- Line 794, “ISO9842: Oil of rose (Rosa x damascena Miller). Available online: (accessed on) please specify.
ISO 9842:2003. International Organization for Standardization: ISO 9842:2003, Oil of rose (Rosa x damascena Miller). https://www.iso.org/standard/28611.html
Table 5 was re-written so that it again includes now only the published literature about R. damascena Mill., Taif rose and R. gallica ‘Francesa’ draft in R. canina. The specific data like IC50 values, percentage of survival, etc on malignant and normal cell lines of the essential oil, different types of extracts, fractions and BAC are not given in the text but only in table in order to avoid repetition of information and to give a better comparative overview on the antineoplastic potential and the relevant in vitro toxicological data about the compared Rosa species. The table consist now of six columns, wherein the anticancer activity is given along with the respective toxicological data and the references are cited in a separate column for each activity. The values indicating promising high activity are denoted with *** and an explanation is given in the legend under the table. As far as no data about the IC50 values on cancer cell lines were found in the scientific databases for R. alba L., R. centifolia L., and R. gallica L., no information concerning the antineoplastic activity of these three species was included in Table 5.
Reviewer 2 Report
Thank you very much for the corrections. My only comment is that the authors instead of reducing the size of manuscript they increased to 46 pages.
Author Response
The authors express their heartfelt gratitude to the esteemed reviewer. In the first version of the manuscript, the tables and figures were attached in separate files. In the second version, tables and figures were combined into one file. For this reason, the new file is larger in size.